# Reducing Outdoor Air Pollutants through a Moss-Based Biotechnological Purification Filter in Kazakhstan

**Andrii Biloshchytskyi** [1,2], **Oleksandr Kuchanskyi** [3], **Yurii Andrashko** [4,*], **Didar Yedilkhan** [5], **Alexandr Neftissov** [5], **Svitlana Biloshchytska** [2,6], **Beibut Amirgaliyev** [5] and **Vladimir Vatskel** [7]

1   University Administration, Astana IT University, Astana 010000, Kazakhstan; bao1978@gmail.com
2   Department of Information Technology, Kyiv National University of Construction and Architecture, 03680 Kyiv, Ukraine; bsvetlana2007@gmail.com
3   Department of Information Systems and Technology, Taras Shevchenko National University of Kyiv, 01601 Kyiv, Ukraine; kuchansky@knu.ua
4   Department of System Analysis and Optimization Theory, Uzhhorod National University, 88000 Uzhhorod, Ukraine
5   Department of Science and Innovation, Astana IT University, Astana 010000, Kazakhstan; d.yedilkhan@astanait.edu.kz (D.Y.); alexandr.neftissov@astanait.edu.kz (A.N.); beibut.amirgaliyev@astanait.edu.kz (B.A.)
6   Department of Computational and Data Science, Astana IT University, Astana 010000, Kazakhstan
7   IT-LYNX LLC, 03194 Kyiv, Ukraine; v.vatskel@it-lynx.com
*   Correspondence: yurii.andrashko@uzhnu.edu.ua

**Abstract:** This study considers the creation of a network of moss-based biotechnological purification filters under the Smart City concept. The extent of the absorption of heavy metals and gases by Sphagnopsida moss under different conditions was investigated. The efficiency of air purification with biotechnological filters was also investigated in the city of Almaty, Republic of Kazakhstan, where an excess of the permissible concentration of harmful substances in the air, according to the WHO air quality guidelines, is recorded throughout the year. Data on the level of pollution recorded with sensors located in the largest Kazakhstani cities from 21 June 2020 to 4 June 2023 were selected as the basis for calculating purification efficiency. In total, there are 220 in 73 settlements of the Republic of Kazakhstan, with 80 such sensors located in the city of Almaty. Since creating a single biotechnological filter is expensive, our task was to calculate the air purification effect in the case of increasing the number of filters placed in polluted areas. We show that 10 filters provide an air purification efficiency of 0.77%, with 100 filters providing an air purification efficiency of 5.72% and 500 filters providing an air purification efficiency of 23.11%. A biotechnological filter for air purification based on moss was designed at Astana IT University by taking into consideration the climatic features, distribution, and types of pollution in the Republic of Kazakhstan. The obtained results are essential for ensuring compliance with the standard for environmental comfort in the Republic of Kazakhstan. Additionally, the research findings and the experience of implementing a moss-based biotechnological filter can be applied to designing similar air purification systems in other cities. This is of great importance for the advancement of the field of urban science.

**Keywords:** biotechnological filter; moss; air pollution; smart city

## 1. Introduction

The problem of air pollution is one of the primary issues posing a severe threat to the health of millions of people. In 2021, the World Health Organization (WHO) estimated that more than 7 million deaths were annually caused by air pollution [1]. The air pollution problem is particularly acute in large cities. This is especially true in those with active industrial facilities or with specific geographical locations that complicate natural air purification. Air pollution not only affects people's health but also significantly worsens

overall quality of life. Both natural and anthropogenic air pollutants can spread over considerable distances and cover large areas via wet and dry precipitation. This poses a severe risk to human health as these pollutants can be inhaled or enter the food chain. There are significant health effects associated with both short-term and long-term exposure to air pollution. Consequently, air quality monitoring and the implementation of engineered solutions for air purification are essential tasks in the field of urban science.

The International Organization for Standardization (ISO) Technical Committee for the Sustainable Development of Communities has developed a new series of international standards to enable an integrated approach to sustainable development [2]. The ISO 37120:2018 [3] (second edition: 2018; first edition: 2014) includes many factors influencing quality of life such as economic factors, education, energy, urban planning, etc. These factors make it possible to measure quality of life, establish the efficiency of city services, and ensure effective urban planning for rapidly growing cities. Research [4] based on the ISO 37120:2018 standard has developed seven dimensions for evaluating urban quality of life. These included urban services, economy, culture and recreation, urban mobility, conviviality, security, and environmental comfort. The latter included noise and air pollution, climate comfort, cleanliness, and wastewater.

In large cities with well-developed transportation infrastructure, the issue of air pollution remains highly relevant. The development of engineering solutions to address this problem is directed towards enhancing the city's sustainability amidst rapid population growth, improving the ecological conditions within the urban environment, and enhancing overall quality of life. The establishment of systematic approaches in order to monitor air pollution levels and formulate long-term strategies for the purpose of enhancing air quality are imperative objectives in urban development. Achieving success in these tasks would result in decreased health risks for citizens, subsequently leading to reduced burdens on city and state budgets from medical care expenses.

The dispersion of different air pollutants follows a non-linear pattern, with concentrations varying across different regions. Consequently, it has become crucial to forecast fluctuations in air pollutant levels and devise engineering solutions to address this challenge. The issue of air pollution holds particular significance for prominent cities and their rapidly expanding agglomerations. A notable example is the city of Astana, the capital of the Republic of Kazakhstan, with a metropolitan population exceeding 1.3 million people. Astana experiences an annual population growth rate ranging from 3.5% to 4%.

The purpose of this study was to investigate methods for reducing environmental pollutants in the Republic of Kazakhstan, specifically by developing and implementing a biotechnological purification filter based on living plants. To achieve this objective, three primary problems needed to be addressed:

- Description of the engineering solution for creating a biotechnological cleaning filter using moss in a way that considers the unique climatic features of the Republic of Kazakhstan;
- Evaluation of the capacity of moss to absorb heavy metals under natural conditions;
- Assessment of the effectiveness of the biotechnological cleaning filter based on moss, taking into account the prevailing level of air pollution.

By applying urban science methodologies to devise engineering solutions that foster a comfortable living environment, cities can become more eco-friendly and sustainable.

## 2. Literature Review

Air pollution presents a multifaceted and intricate challenge, demanding comprehensive solutions. It is imperative that researchers adopt a multifaceted approach incorporating the monitoring and forecasting of air pollution levels across different areas of cities. This would facilitate the analysis of pollution sources and enable policymakers to propose corrective actions to local authorities. The establishment of air quality standards and systematic assessment and management measures are simultaneously vital components of an effective solution. The development of a long-term strategy for ensuring citywide air

quality is an integral part of urban science. Large cities, characterized by dense populations and increased health risks, face particularly acute air pollution issues. Currently, more than 50% of the global population resides in urban areas. A noteworthy study [5] explored the correlation between air pollution and the risk of emergency care for young children with asthma. Such research has emphasized the critical nature of air purification efforts in ensuring the high quality and safety of people's lives.

A current trend in addressing air purification challenges involves the development of biotechnological filter systems that utilize specific plant crops to remove pollutants while releasing oxygen. This approach has gained traction, with relevant projects securing government funding in countries such as Great Britain, Germany, Portugal, and the Republic of Kazakhstan [6]. These filters offer the advantage of flexibility in placement, as they can be deployed in various locations within a city. However, their cost remains a hindrance, and they are yet to be mass-produced. Consequently, it has become crucial to judiciously employ these filters, situating them solely in areas where their use is warranted. In [7], a model was constructed to identify the optimal placement of biotechnological filter systems for air purification by addressing the discrete optimization problem. This model provided forecasts of the air quality index for a specific region, ensuring the efficient deployment of these biotechnological systems to improve air quality.

Various engineering solutions are being developed for this purpose. For example, the concept of the CityTree vertical plant filter was described in [8]. This comprises a vertical plant structure that not only cleans the air but also cools the environment, retains water, and reduces noise. According to the findings in [6], the innovative CityTree device effectively reduced air pollution from harmful solid particles and gases, including $NO^2$ and $CO^2$. Remarkably, one such device can replace the equivalent of 275 trees, while requiring minimal urban area for placement. However, the application of such filters in the conditions of Kazakhstan poses challenges due to its predominantly continental climate zone. Astana, in particular, globally ranks as the second coldest capital city. The utilization of open purification biotechnological filters in these conditions has proved ineffective as plants require the maintenance of stable temperatures and humidity throughout the growing season.

Moss is an intriguing plant known for its capacity to achieve a high level of air purification, particularly under conditions of stable temperature and humidity. Previous research [9] outlined a moss biomonitoring method utilizing three moss types: Pleurozium schreberi, Sphagnum fallax, and Dicranum polysetum. This method involved measuring the concentration of analytes (Mn, Fe, Cu, Zn, Cd, Hg, and Pb) accumulated within the total suspended particulate matter collected from dust collector filters in Opole (Opole Voivodeship, Poland). Engineering solutions for developing a moss-based air purification system have been described in other studies [10] that have detailed a biotechnological prototype capable of exercising automated control over moss growth and productivity. Consequently, stable air purification efficiency can be ensured once the filter is installed. However, if researchers are to evaluate a filter's effectiveness, it is necessary to collect enough data from air-quality-monitoring sensors both before and after the filter's installation, while also accounting for the costs of ensuring the uninterrupted operation of the filter and other relevant factors.

An essential component of the filter's effective operation is the provision of constant air quality monitoring. As demonstrated in the study [11], air pollutants in a city exhibited uneven distribution in both space and time. Consequently, certain regions experience the regular exceedance of permissible pollutant concentrations, posing health risks to city dwellers. In [12], researchers constructed a multi-criteria model for the optimal placement of green areas in the city based on demographic indicators. However, this model did not account for the dynamics of city development. In contrast, the study [13] formulated a multi-criteria placement optimization model aimed at maximizing air purification while minimizing the heating of street surfaces. This proposed model enabled the selection of the most suitable option from the predefined plans for green areas. The outcomes of

solving these problems can contribute to the optimal placement of biotechnological filters throughout a city.

The city of Astana, which serves as the capital of the Republic of Kazakhstan, sustains a population of over 1.3 million individuals. As of 2023, it is expected to maintain a moderate level of air quality, as denoted by an AQI (Air Quality Index) score of 53. During the period from 14 June to 14 July 2023, an average AQI score of 39 was recorded [14], aligning with the latest WHO air quality guidelines [15]. The principal sources of urban air pollution in Astana encompass stationary thermal power plants, boiler houses, vehicular emissions, construction sites, production facilities, cement, and asphalt factories. Past data indicated a declining trend in pollutants emanating from stationary sources in the city [16]. However, this trend remains vulnerable to rapid changes due to the city's burgeoning growth, the continuous increase in the number of vehicles, and the presence and use of older car models, the emissions of which surpass acceptable environmental safety levels. Furthermore, the air quality in Astana experiences significant deterioration when the air temperature drops. This is particularly true during the onset of winter and periods of low wind speed, as these conditions foster the accumulation of small polluting particles.

The tendency towards air quality deterioration is a recurring issue observed in certain regions of Kazakhstan, notably in the largest city, Almaty, situated in the southern part of the country. Almaty exhibits PM2.5 concentration levels 1.5 times higher than those specified by the WHO's annual air quality guideline value [17], which can largely be attributed to the city's specific geographical location. Resolving this complex environmental situation necessitates engineering solutions, particularly the development of specialized filters designed to enhance air quality.

To address air quality concerns, Kazakhstan has implemented systemic measures at both the state and local levels. For instance, in accordance with the Paris Agreement on climate change, the Republic of Kazakhstan aims to reduce carbon dioxide emissions by 45 percent by 2030 and to achieve zero emissions by 2050. As part of this commitment, Astana plans to gradually replace all public transportation with electric buses. Additionally, efforts have been made to restrict large-sized vehicles and heavy-duty equipment in the central part of the city, alongside the establishment of a transparent permit system. Moreover, periodic inspections ensure compliance with environmental requirements when using vehicles. As a pressing objective, the development of a biotechnological filter specifically tailored to effectively purifying the air in a particular region is underway. A key aspect of the research involves evaluating the efficiency of such filters in removing specific pollutants from the air.

### 3. Materials and Methods

*3.1. Basic Concepts: Creation of a Moss-Based Biotechnological Purification Filter, Taking into Account the Climatic Features of the Republic of Kazakhstan*

An essential consideration in the development of a biotechnological purification filter based on moss for installation in Kazakhstan's conditions is the country's predominant location in the continental climate zone. In such climatic conditions, many plants struggle to survive. Consequently, it has become imperative to create a closed system in order to ensure the effective operation of the filter.

The following technical characteristics were used during the construction studies for the biotechnological filters:

1. Frame size: 2.8 m (height) × 2 m (width) × 2 m (length).
2. The entire frame structure was made to be partially collapsible and convenient for transportation. Effective thermal insulation was provided within certain standards to avoid heat loss in the structure. The frame was resistant to strong winds, which are observed in most of Kazakhstan during a certain period of the year. The metal structure of the frame of the biotechnological filter was made of non-corrosive materials. All connections were tested for strength and tightness.

3.  In the manufacturing of the external frame (front and side), profiles, and corner connectors, aluminum was used as the primary material, namely, aluminum sheet 2 mm "Alkan" and aluminum partitions, which were assembled and then painted with polymer paint (color RAL 7035).
4.  The upper and lower parts of the filter were made of sheet metal with a thickness of at least 2 mm. The upper and lower parts were insulated with special extruded insulation with a thickness of at least 40 mm. Crossbar reinforcement was provided in order to install solar panels on the upper part. The crossbars were installed every 50 cm. In the lower part of a crossbar, there was a hole for connecting communications.
5.  The front sides were made of energy-efficient double-glazed windows filled with argon using tempered glass on all sides. The sides were divided into four parts using dividers made of aluminum profiles. There was a glass thickness of at least 4 mm.
6.  The sides included a cavity for installing a demo screen (a 43-inch monitor) covered by protective glass. The slot for the monitor was insulated and did not allow cold and wind to pass through. Under the monitor, through-holes were provided to organize the controlled entry and exit of air from two sides. The side was removable to allow for system maintenance and plant access.
7.  The inner part of the frame provided racks measuring 1 × 1 m in order to install and fasten filter components. There was a bench around the 360-degree biotech filter that was at least 45 cm high, at least 45 cm deep, and made of wood. The design of the bench was mobile.

After the construction work was completed, the appearance of the biotechnological filter looked as shown in Figure 1.

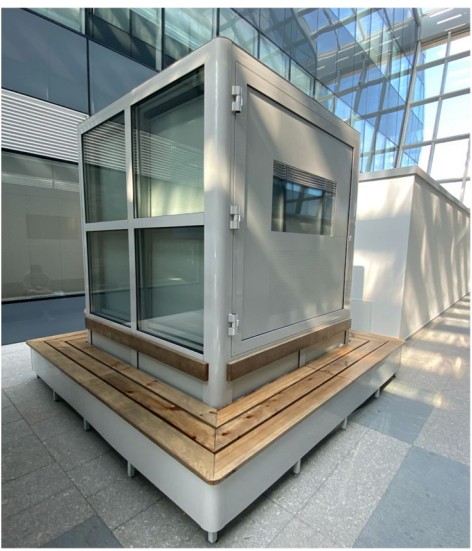

**Figure 1.** The appearance of the biotechnological filter installed in the hall of Astana IT University.

Figure 2 shows a biotechnological system in the middle of the filter. The installation consisted of eight vertical trays on two opposite sides, four on each. The size of the vertical tray was 95 × 90 cm. Six horizontal trays were installed between the vertical trays. The size of the horizontal tray was 80 × 80 cm. The trays were made of stainless material and painted with polymer paint. The total height of the farm was 270 cm, and the occupied area was 3.24 m². The culture was placed in trays on top of the selected substrate. A net was used to fix the moss and substrate during vertical cultivation.

A reservoir was installed under the horizontal trays to irrigate the culture. The volume of the tank was 0.32 m³. A water sprinkler was installed above each horizontal tray. Galvanized sheets were mounted around the tray's perimeter to insulate against water splashes. When vertically growing moss, holes for the irrigation of crops were provided above each tray. Water was supplied to the sprayer using a pump. Trays, both vertical and

horizontal, could slide out for convenient operation. All conditions (water rate, humidity, temperature, and lighting) for crop cultivation were controlled using a controller and a minicomputer.

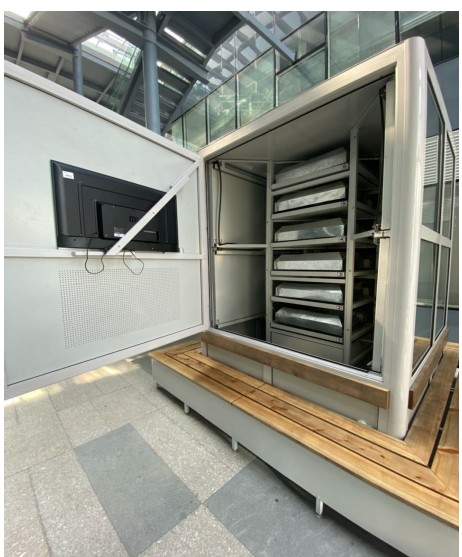

**Figure 2.** A biotechnological system (farm) located in the middle of the filter.

Figure 3a,b show an experimental setup of the moss trays at a biotech filter farm.

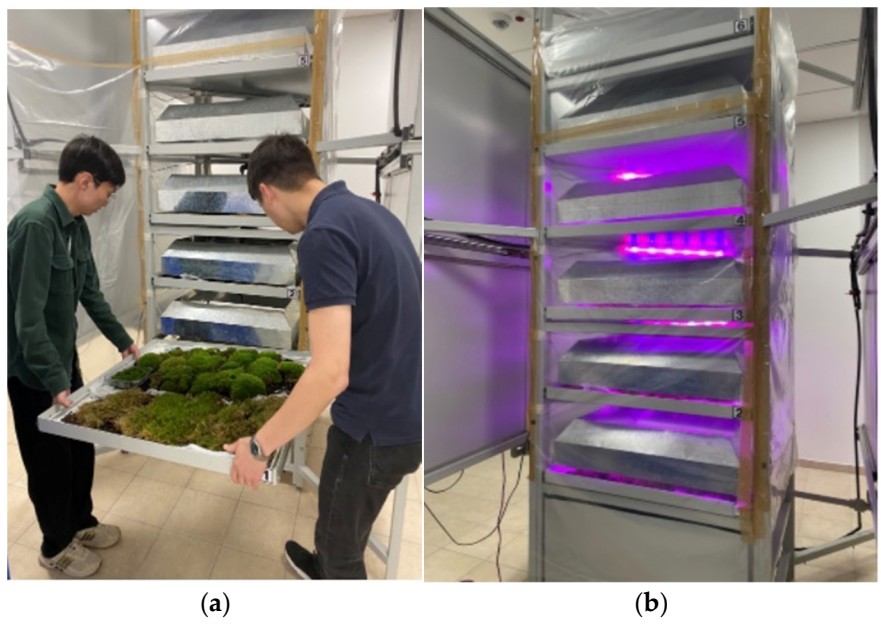

(**a**)                                (**b**)

**Figure 3.** Experimental setup of the moss farm. Note: (**a**) planting the moss to the biofilter; (**b**) artificial lighting of the moss in the biofilter.

A set of FGHGF sensors with a maximum measurement temperature of 49 degrees Celsius was used to monitor the temperature and humidity in the filter. A MAG-6 gas analyzer was used to perform the air quality analysis.

### 3.2. Automated Microclimate Support System in the Biotechnological Air Purification Filter

Figure 4 shows that the universal programmable logic controller (1) had input and output channels connected to intermediate relays. In turn, the light sensor (2), air temperature and humidity sensor (3), and sawdust meter PM 2.5–10 (4) were connected to the input channels of controller 1, the output channels of which were connected through intermediate

relays to the cooling system/air conditioner (5), heat generator/heating system (6), air humidifier (7), LED lamp 8, air pumping fan (9) and drip irrigation pump (10). A universal controller based on a Cortex A57 was used as a programmable log controller (1), a GY-30 (BH1750FVI, I2C) as a light sensor (2), a Temperature and Humidity Transmitter FGHGF as a temperature and humidity sensor (3), a PMS5003 high-precision laser sensor pm2.5 as a sawdust meter (4), a Midea MPPDA09CRN7 as a cooling/conditioning system (5), a Ballu B heating system IH-LW-1.2 as a heat generator (6), a Deerma DEM-F500 as a humidifier (7), FitoLED 20 bicolor equipment as lighting lamps (8), a channel fan with three speeds as an air pumping fan (9), and a Magnetta 1AWZB550 as a drip irrigation pump (10).

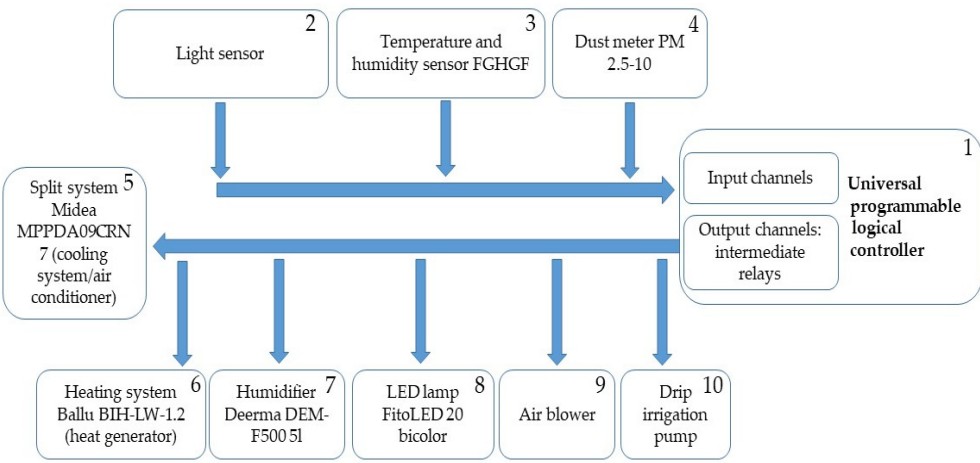

**Figure 4.** Structural model of the automated microclimate support system in the biotechnological air purification filter.

The conceptual model of the data exchange and management of the biotechnological filter is presented in Figure 5.

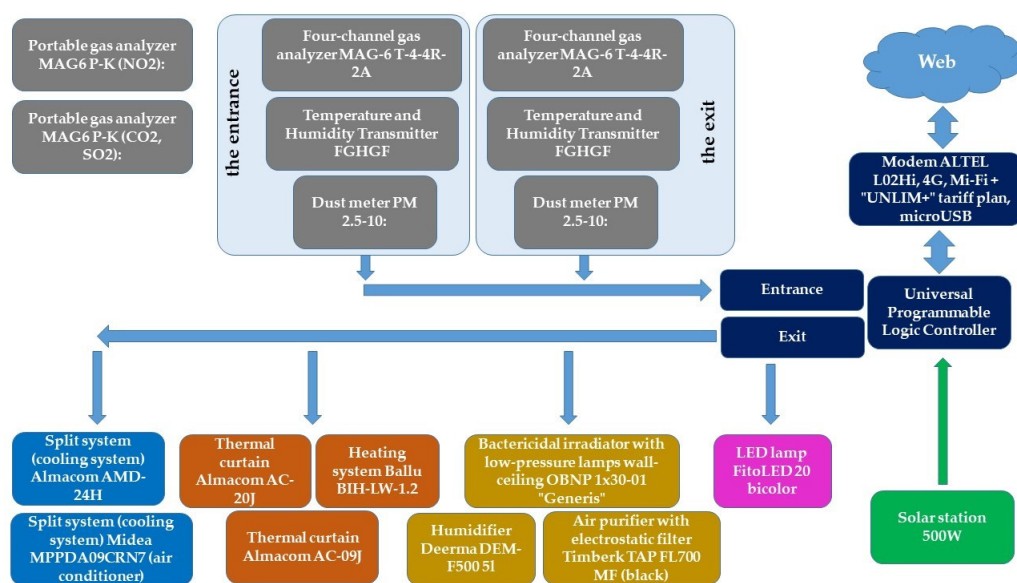

**Figure 5.** Conceptual model of the data exchange and management of a biotechnological filter.

The universal programmable logic controller (1) was installed inside the biotechnological filter and connected to the power supply. After being switched on, the controller (1) turned on the air pumping fan (9) through the filter with the specified rotation frequency n and interrogated the light sensor (2), air temperature and humidity sensor (3), and PM 2.5–10 dust meter (4) connected through the input channels. After interrogating and receiving data from the sensors, the controller compared the results with the programmed

values of the settings and values of the parameters required to support the necessary microclimate inside the biotechnological filter. When the polled value from the sensors was different than the set value of the parameter, controller (1) gave a command to the output channels through the intermediate relays to turn on or off the connected cooling/air conditioning system (5), heat generator/heating system (6), air humidifier (7), and LED lamp (8). After an interval, it polled the data from the sensors again and, if necessary, exerted a control effect through intermediate relays. After a specific interval of t2, the drip irrigation pump was turned on. It was impossible to maintain the parameters of temperature, humidity, and dust inside the biotechnological filter at the initial fan rotation speed of 9 n. As such, the controller reduced the speed by the value m. After a specific time, interval t3, controller (1) sent a signal to turn on pump (10) to water the integrated moss modules. Thus, inside the biotechnological filter, the constant parameters of the necessary microclimate were maintained to ensure the functioning of integrated moss modules tasked with air purification.

## 4. Results

### 4.1. Collection of Data

As biotechnological filters entail considerable expenses, a prudent approach is crucial in selecting appropriate locations for their placement within a city. The filters should be strategically positioned where they can exert a significant impact on air purification. To identify the most rational points for filter placement, a network of 220 sensors was considered across 73 cities within the Republic of Kazakhstan (Figure 6). Table 1 provides a breakdown of the number of sensors in the selected cities. In the remaining cities of Kazakhstan, four or fewer sensors have been installed. Detailed information on the location of the sensors can be found in Supplementary Table S1. The most significant number of sensors was installed in Almaty because this city traditionally observes an excess of pollution indicators. This is primarily due to the city's geographical location and dense population [18].

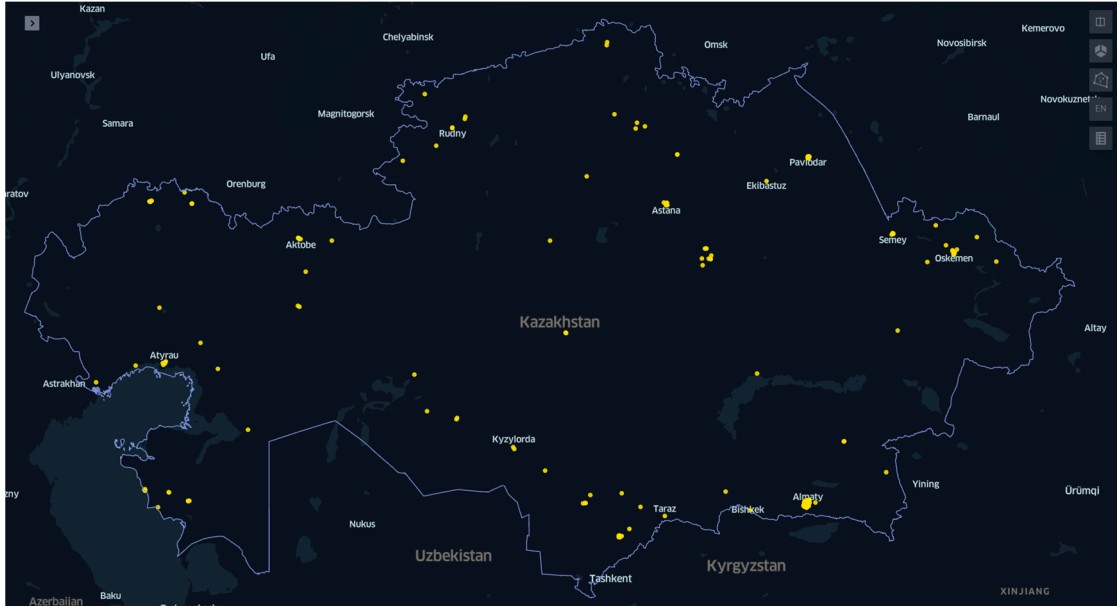

**Figure 6.** Location of air pollution monitoring stations in the Republic of Kazakhstan (made using the Kepler service).

The sensors record the content of CO, $H_2S$, NO, $NO^2$, $SO^2$, dust in general, and $PM_{2.5}$ and $PM_{10}$ in the air every hour. Indicators were collected hourly for the period from 21 June 2020 to 4 June 2023. The sensors do not have a means of detecting the concentration of heavy metals in the air. However, the study [19] estimated the concentration of heavy

metals based on the amount of PM$_{2.5}$. Additionally, the corresponding coefficients, which are listed in Table 2, were calculated using Formula (1), where C$_i$ is the concentration of the i-th heavy metals for i = 1,9, k$_i$ is the ratio coefficients, and C$_{2.5}$ is the concentration of the PM2.5 air pollution.

**Table 1.** Number of sensors.

| No. | City | Number of Sensors | Population |
|---|---|---|---|
| 1 | Almaty | 80 | 2,147,233 |
| 2 | Pavlodar | 12 | 335,272 |
| 3 | Oskemen | 10 | 326,498 |
| 4 | Astana | 9 | 1,350,228 |
| 5 | Aktobe | 6 | 413,918 |
| 6 | Atyrau | 5 | 235,314 |
| 7 | Shymkent | 5 | 932,235 |

**Table 2.** Coefficients for the conversion of PM$_{2.5}$ concentration into the corresponding metal concentrations.

| | Coefficients, µg/mg | | | | | | | | |
|---|---|---|---|---|---|---|---|---|---|
| | **Pb** | **Cd** | **Zn** | **Cu** | **Fe** | **Ni** | **Co** | **Mn** | **Cr** |
| Min | 0.07 | 0.01 | 0.20 | 0.02 | 0.86 | 0.07 | 0.01 | 0.01 | 0.04 |
| Max | 0.30 | 0.01 | 0.75 | 0.14 | 2.02 | 0.30 | 0.01 | 0.01 | 0.12 |
| Average | 0.17 | 0.01 | 0.45 | 0.05 | 1.50 | 0.17 | 0.01 | 0.01 | 0.08 |

During data collection, 1760 time series were obtained, totaling 12,785,943 points. Detailed sensor data can be found in Supplementary Table S2.

*4.2. Evaluation of the Degree of Absorption of Heavy Metals by Moss in a Biotechnological Purification Filter*

Solving the task of creating the optimal conditions for the growth of plants responsible for air purification is an integral component of a biotechnological cleaning filter. For this study's prototype of the biotechnological filter, a plant—moss (in the taxonomic division Bryophyta), in particular, Sphagnopsida—was chosen.

The use of Sphagnopsida helps to solve the problems of biomonitoring environmental pollution and CO$^2$-neutral agriculture in order to combat climate change [20]. Systematic moisture provision is a mandatory condition for growing Sphagnopsida, as the plant cannot actively control the water supply system [21]. Sphagnum can absorb an amount of water that exceeds 20 times its weight. Sphagnum has no roots or rhizoids. Only the upper part grows, and stems are formed. The lower part of the stems dies and transforms into peat. Most species of sphagnum are adapted to conditions of low light [22].

During the study, the temperature regime for sphagnum was found to be 3–22 °C in vivo or 10–20 °C in vitro. During the day, the optimal air temperature for growing sphagnum in a closed system is 22 ± 1 °C. At night, it is 16 ± 1 °C. The requisite light period is 16 h, and the relative humidity is 85 ± 15%. The in vitro temperature was chosen as a compromise between the recommendations of biologists for optimal temperatures for moss growth and the minimization of energy consumption.

From 28 September to 5 October 2022, an experiment was conducted to determine air quality in terms of the presence of heavy metals in the city of Astana (Republic of Kazakhstan). Sphagnum moss bags were suspended on the 2nd floor of a residential complex (position 1) and in the biofilter (position 2) to determine the degree of the accumulation of heavy metals. At the end of the experiment, moss samples were sent to specialists. In the laboratories "Kazecoanalysis" in the city of Astana, the concentrations of heavy

metals were determined, and relevant acts were issued with the results of the concentration measurements. The air temperature during the experiment for position 1 averaged 20–21 degrees Celsius and the humidity was −54–58%. The results are given in Table 3.

**Table 3.** The concentration of heavy metals in selected samples of Sphagnopsida moss from Astana.

| The Location of the Moss | Metals, mg/kg | | | | | | | | |
|---|---|---|---|---|---|---|---|---|---|
| | **Pb** | **Cd** | **Zn** | **Cu** | **Fe** | **Ni** | **Co** | **Mn** | **Cr** |
| Position 1 | 4.0 | <0.05 | 31.3 | 5.9 | 699 | 1.7 | <0.6 | 60.8 | 0.4 |
| Position 2 | 3.0 | <0.05 | 28.9 | 4.9 | 625 | 1.5 | <0.6 | 48.1 | 0.4 |

On the basis of the results of the experiment, it can be concluded that Sphagnopsida moss absorbed heavy metals to enable its use as the basis of biotechnological purification filters. The conducted experiment showed that Sphagnopsida absorbed heavy metals with the same efficiency in both the open air and in the middle of the biotechnological filter. The stability of the temperature regime did not significantly increase the absorption of heavy metals. The obtained results made it possible to focus on ensuring the survival of moss in a biotechnological filter.

*4.3. Evaluation of the Efficiency of the Biotechnological Cleaning Filter Based on Moss, Taking into Account the Level of Air Pollution*

The task of evaluating the effectiveness of the biotechnological cleaning filter based on moss in relation to the level of air pollution was divided into two subtasks. The first subtask was to maintain stable conditions in the filter. Maintaining a stable temperature and air humidity in the middle of the filter is essential. This is necessary in order to grow moss, the primary component of air purification. A control model was used, which was described in more detail in the study [10]. Figure 7 shows a conceptual model of the system functioning for monitoring stable conditions in the middle of the filter. This model contained vector s, which described the parameters essential for the functioning of the filter. These parameters included air temperature, humidity, and pollution, characterized by the concentration of harmful substances in the air. The state vector additionally included time.

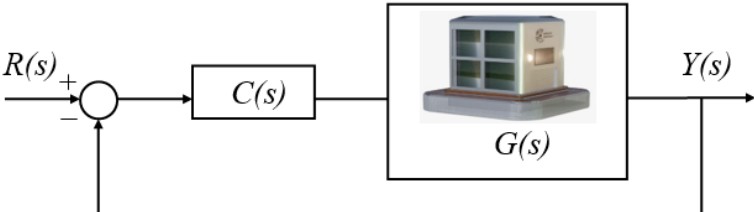

**Figure 7.** Conceptual model of the functioning of the control system of stable conditions in the filter.

The function R(s) determined the state of the environment in the location where the moss-based biotechnological purification filter was functioning. The function R(s) value could be calculated based on the data collected during air condition monitoring.

The control function C(s) determined the actions to be taken before the supply air was provided to the moss chamber in the middle of the filter. As a result of such preparation, the temperature and humidity of the air changed. A previous study [10] showed that the function C(s) can be found using Equation (1).

$$C(s) = k\left(1 + \frac{1}{T_i s}\right)G^{-1}(0) \tag{1}$$

Calculating the control function was one of the tasks solved with the help of software that supported the operation of the biotechnological filter.

Function G(s) determined the performance of a biotechnological purification filter based on moss. The principal expected result was a decrease in the concentration of pollutants in the air due to their absorption by moss. A linear approximation for the function can be calculated based on the research conducted.

$$G(s) = \Gamma \times s \tag{2}$$

where $\Gamma$ is the transformation matrix, the coefficients of which are empirically calculated. In future studies, we will assume that this matrix is diagonal for the purpose of simplification. That is, the absorption of each pollutant depends only on its concentration and does not depend on other parameters.

The function Y(s) determined the parameters of the air coming from the biotechnological purification filter into the environment.

The second subtask consisted in evaluating the effect of filter placement. In order to solve this sub-problem, a spatial model was built to assess the impact of the placement of biotechnological filters on air quality. Considering the collected data from air quality monitoring in the Republic of Kazakhstan, the construction of the model was carried out for the city, about which the maximum data are known, namely, for the city of Almaty. The task of the optimal placement of filters was described in [7].

The spatial model for assessing the impact of the placement of biotechnological filters on air quality contained several simplifications. In particular, the city was covered with a hexagonal grid. The state of the air parameters was assumed to be the same for the entire sub-region bound by the hexagon. Given that the air condition was known only for some subregions, the air condition in other subregions was calculated as the arithmetic average of the air conditions of neighboring subregions. Let a hexagonal grid with n hexagons cover the city; $O = (o_1, o_2, \ldots, o_n)$. Then $s_i$, $i = \overline{1,n}$ is the state of the air in the subregion bounded by the hexagon $o_i$. Let us mark all neighboring hexagons to the hexagon $o_i$ as $(o_{i+1}, o_{i+2}, \ldots, o_{i+m})$, $o_j \in O$, $j = \overline{i+1, i+m}$, and $m \leq 6$. Then, the state of the air in the hexagon $s_i$ is defined as:

$$s_i = \frac{1}{m}\sum_{j=1}^{m} s_{i+j} \tag{3}$$

This method of assessing the air's condition was applied to cells that lacked the requisite sensors to determine air pollution levels. In this study, the efficacy of biotechnological filters for air purification in urban areas was evaluated by considering their quantities based on the size of the hexagonal grid cells. The utilization of such a model resulted in a system of large-scale linear algebraic equations characterized by a highly sparse matrix. The number of unknowns in the system corresponded to the number of subregions, and the number of coefficients different from zero in each row did not exceed seven. The model assumed the installation of one biotechnological purification filter in specific subregions. For the simplicity of simulation, we considered all biotechnological purification filters to be identical, to operate continuously, and to exert a linear influence on the air condition within the subregion.

The city of Almaty was divided into a grid comprising 3447 hexagons (Figure 8). Within the city, 80 sensors were strategically placed to monitor the level of air pollution, and these sensors collected air condition data from 21 June 2020 to 4 June 2023. Prior to the installation of biotechnological filters based on moss, the air quality within the city was simulated. The air condition in 80 subregions, delineated by the corresponding hexagons, was determined using data from the installed sensors. For the remaining 3367 subregions, the air condition was computed using Formula (2). This process entailed hourly calculations for the specified period. Figure 9 presents a map illustrating the average pollution levels in Almaty.

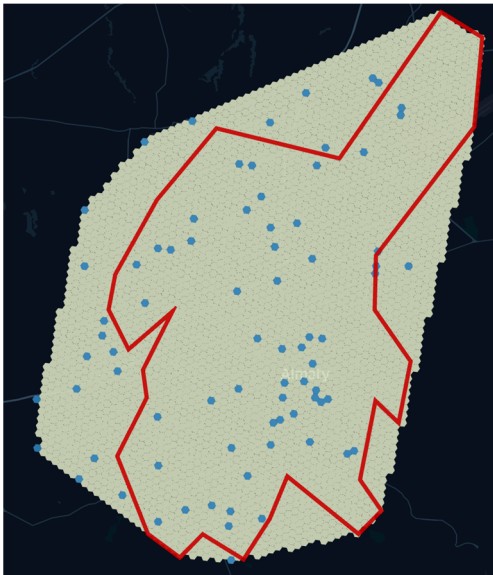

**Figure 8.** Covering the city of Almaty with a hexagonal grid.

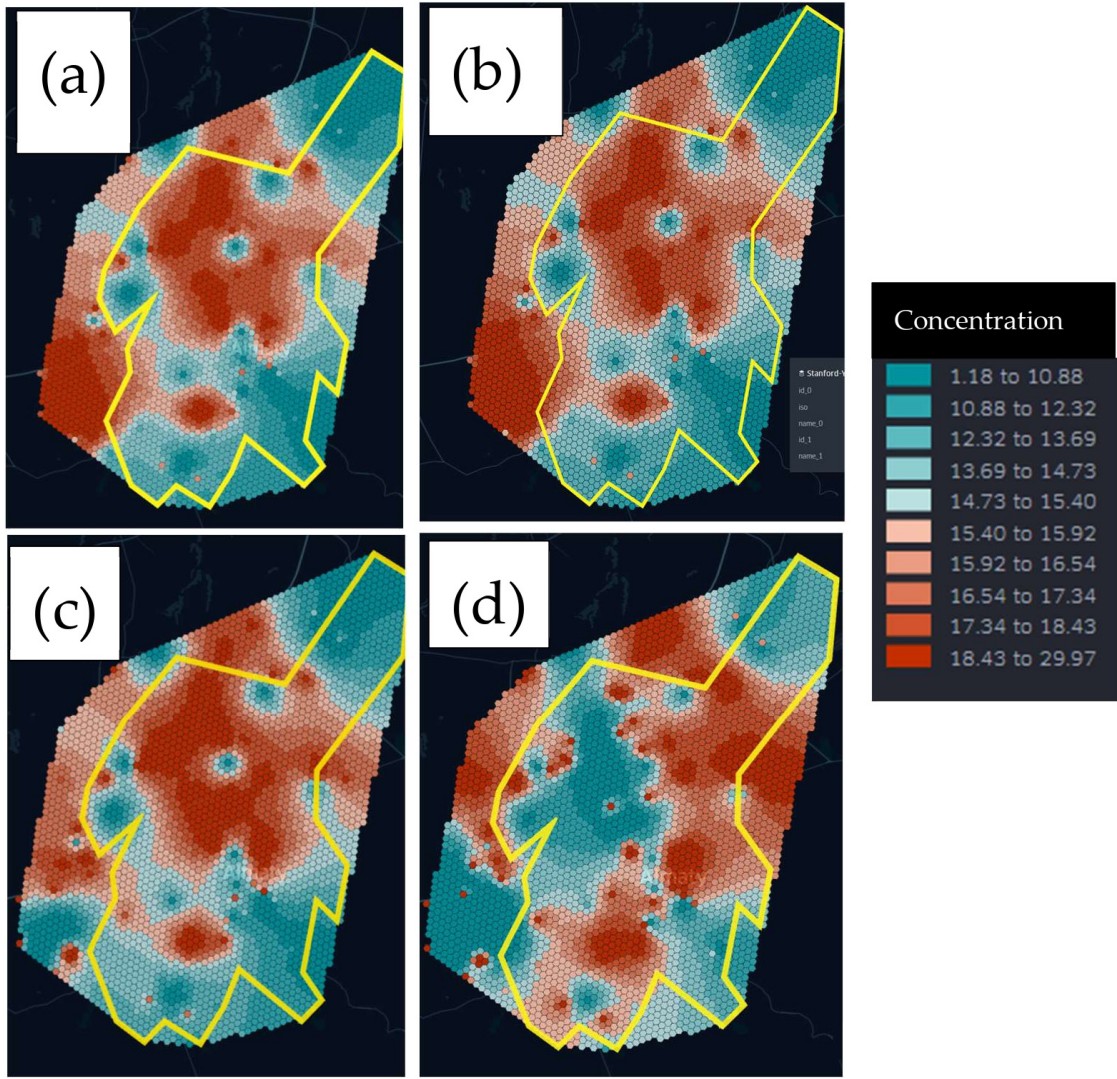

**Figure 9.** Map of the concentration of the Pb pollutant in the air of Almaty when using (**a**) 0 filters, (**b**) 10 filters, (**c**) 100 filters, and (**d**) 500 filters.

In order to assess the effectiveness of air purification using the filters, an incremental approach was adopted, wherein one filter, two filters, and so forth, were placed in the most polluted regions. For each scenario, air quality indicators were computed using a spatial model.

Descriptive characteristics pertaining to pollution were then derived from the spatial model, enabling a comparison of the effects before and after the installation of the filters. Employing the coefficients specified in Table 2, the concentration of heavy metals was determined based on the concentration of PM2.5. As a result, the model provided proportional indicators for all heavy metals. Subsequent calculations focused solely on one specific contaminant, namely, lead (Pb). Given the limitations of the proposed methodology, the results for all pollutants were proportional. The problem of biofilter locations for all pollutants have the same solution. As shown in the study [23], Pb was one of the most dangerous pollutants that adversely affected the health of adults and children. In Kazakhstan, Pb is one of the air pollutants with a high concentration.

When building the model, we used the parameters of the absorption of heavy metals by the biofilter, which were experimentally obtained. Moreover, in the modeling, an assumption was made that the mass of the moss in the biofilter did not change, and that the moss uniformly absorbed heavy metals. The concentration of heavy metals in the air was calculated according to Formula (1), the coefficients for which were taken from the study [19]. The calculations assumed that one biotechnological purification filter contained 100 kg of Sphagnopsida moss, which, according to Table 3, absorbed 300 mg of lead every week. The calculation of the efficiency of placing a certain number of filters was based on comparing the difference with the pollution of the model without filters. The maximum difference, mean difference, and relative efficiency were calculated. The results are shown in Table 4.

**Table 4.** The effect of the use of biotechnological purification filters in the city of Astana to purify the air from lead.

| Concentration Pb, $\mu g/m^3$ | Number of Filters | | | |
|---|---|---|---|---|
| | **0** | **10** | **100** | **1000** |
| Max | 29.966 | 28.266 | 28.067 | 25.935 |
| Min | 1.178 | 1.178 | 1.178 | 0.437 |
| Average | 15.003 | 14.885 | 14.143 | 11.532 |
| Median | 15.400 | 15.303 | 14.595 | 12.149 |
| Variance | 2.809 | 2.705 | 2.398 | 3.234 |
| Maximum change | 0.000 | 1.928 | 12.435 | 23.682 |
| Average change | 0.000 | 0.116 | 0.859 | 3.470 |
| Efficiency | 0 | 0.77% | 5.72% | 23.11% |

## 5. Discussion

### 5.1. Findings

The simulation results revealed that the application of this air purification concept in the city requires careful consideration of the number of filters needed to achieve the desired cleaning effect. However, given the expensive nature of the filters, it becomes imperative to allocate suitable locations for their installation. Completely cleaning the city of all pollutants would necessitate the installation of tens of thousands of filters, which would be impractical. Moreover, an increase in the number of filters may disrupt the linearity of pollutant absorption, rendering the model inadequate to the task of describing the purification process. Therefore, achieving a high level of urban quality of life demands a comprehensive approach to assessing environmental cleanliness.

The obtained results will enable the strategic planning of filter placement and the determination of the required quantity to achieve sufficient air purification in cities. Consequently, it will become possible to effectively address pollution levels in cities, aligning them with the air quality guidelines set forth by the WHO.

*5.2. Limitations and Future Research Lines*

Assumptions and simplifications were employed during the modeling process. Notably, the absorption of pollution by moss was assumed to linearly occur. However, other research [9] has suggested that the absorption of polluting substances by moss generally follows a non-linear pattern. The model calculated the spread of pollution, assuming that pollution levels would be averaged out in neighboring subregions defined by a hexagonal grid. It is essential to recognize that the dispersion of environmental pollutants is a complex process influenced by various factors, including geography, wind direction, air humidity, and more [24].

Future research should aim to establish the long-term ecological impact of pollution in Kazakhstan's cities. Additionally, scholars should evaluate the economic effects and urban quality of life based on survey data from residents and quantitative indicators of air quality.

In this study, the placement of one filter in each hexagon of the grid was considered. However, it is necessary to explore options such as combining filters or placing multiple filters in hexagons with high levels of air pollution to ensure enhanced air purification efficiency.

The simulation involved calculating the effect of placing more than 800 filters in Almaty. As the number of filters increased, some hexagons displayed negative pollutant concentration values. In such cases, it would be advisable to develop an alternative model to evaluate the effectiveness of air purification based on biotechnological filters.

Currently, the developed biotechnological filter based on moss is undergoing placement on a street in Astana outside the university laboratory. Experimental data on the filter's effectiveness in natural conditions will be recorded shortly.

**6. Conclusions**

This article outlines the development of a biotechnological purification filter based on moss as part of the implementation of the Smart City concept in Kazakhstan. The engineering solution for outdoor filter use was comprehensively described, taking into account the climatic peculiarities of the majority of the Republic of Kazakhstan, especially its continental climate. The core of the biotechnological filter consisted of a farm with Sphagnopsida moss, and the system automatically regulated temperature, watering levels, and air humidity in order to facilitate moss growth. The investigation focused on the absorption capacity of Sphagnopsida moss for heavy metals under various conditions, demonstrating the viability of using this moss for air purification in natural environments. To evaluate the effectiveness of air purification with a biotechnological filter based on moss, modeling was conducted in the city of Almaty, Kazakhstan. This city recorded persistent exceedances of harmful substance concentrations in the air, measured according to WHO air quality guidelines, spanning almost the entire year. Data collected with sensors scattered throughout the Republic of Kazakhstan from 21 June 2020 to 4 June 2023 were employed as the basis for the efficiency evaluation. A total of 220 sensors were deployed across 73 settlements in Kazakhstan, with 80 sensors dedicated to Almaty. The results demonstrated that 10 filters achieved an air purification efficiency of 0.77%, while 100 filters yielded 5.72%, and 500 filters attained 23.11% efficiency.

The design of the biotechnological filter for air purification, based on moss, was executed at Astana IT University, considering the specific climate, pollution distribution, and types prevalent in Kazakhstan. These findings hold significant importance in terms of attaining compliance with the ISO 37120:2018 standard concerning environmental comfort, particularly in the Republic of Kazakhstan.

Since the cost of designing, developing, and installing such biotechnological filters is high, calculating air purification efficiency depending on the number of installed filters is essential for local and state budgets. The obtained research results and the experience of implementing a moss-based biotechnological filter in the conditions of the Republic of Kazakhstan can be applied to the design of similar systems in other countries.

**Supplementary Materials:** The following supporting information can be downloaded at: https://www.mdpi.com/article/10.3390/urbansci7040104/s1. Table S1: Sensor's locations; Table S2: Air pollution measurements.

**Author Contributions:** Conceptualization and methodology, A.B., Y.A., D.Y. and O.K.; software, Y.A.; analysis, O.K., Y.A., A.B., B.A. and D.Y.; coding, Y.A.; writing—original draft preparation, O.K. and Y.A.; writing—review and editing, A.B., S.B., V.V. and. A.N.; visualization Y.A.; project administration, A.B. All authors have read and agreed to the published version of the manuscript.

**Funding:** This research was funded by the Committee of Science of the Ministry of Science and Higher Education of the Republic of Kazakhstan (Grant No. BR10965311 "Development of the intelligent information and telecommunication systems for municipal infrastructure: transport, environment, energy, and data analytics in the concept of Smart City").

**Data Availability Statement:** All data are available in this publication.

**Acknowledgments:** The authors thank the reviewers and editors for their generous and constructive comments that have improved this paper.

**Conflicts of Interest:** The authors declare no conflict of interest.

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
