# Peer review of "Reducing Outdoor Air Pollutants through a Moss-Based Biotechnological Purification Filter in Kazakhstan"

_urbansci, doi:10.3390/urbansci7040104_

Round 1
Reviewer 1 Report
Point 1:
You mast measure the concentration of heave metals, so you must not estimate the concentration of heave metals.
Point 2
The whole manuscript is very difficult to understand, for example
Author Response
The measurement results were carried out in laboratory conditions for 2 samples in the table 3. We do not have sufficient technical means to measure the concentration of heavy metals in the cities of the Republic of Kazakhstan, so we conducted modeling of air pollution and determining the positive impact of biofilters based on estimates for pollution.
Before printing our research, MDPI's English language proofreading service will be ordered and deficiencies will be corrected for better understanding of the article.
Reviewer 2 Report
This is a very interesting paper which describes the development of moss-based purification device for cleaning air and the simulated effects of this novel device installed in Kazakhstan. Even though the device does not have enough effect on the improvement of air pollution in Kazakhstan, this trial is worth publishing when properly revised.
The main problem of this manuscript is the lack of enough data on the performance of the moss-based purification device. First, a more detailed description of the experimental method is required in Section 4.2 (What is residential complex and biofilter? How did you determine concentrations of metals in the samples? How were the blank concentrations (moss filter without exposure to air pollutants) and so on.) Second, it was not clearly described that the link between experimental results on the filter performance and the following simulation. How were the experimental results considered in the parameters of the simulation. Third, why did authors focus on lead (Pb) in the simulation?
Minor points
Table 4  Dispersion → Variance
Figure 9(a)-(d) It is better that all figures should be the same size.
Author Response
Thank you for your valid comments, which significantly improved our research.
1.1 Position 1 is located inside the premises of the university, position 2 is located in a biofilter - a special technical device that maintains constant temperature and humidity. The prototype of the biofilter is shown in Figures 1-3.
1.2. At the end of the experiment, the moss samples were sent to specialists. Laboratories in the city of Astana, where concentrations of heavy metals were determined and relevant acts were issued with the results of concentration measurements.
1.3. blank concentrations of moss without exposure to air pollutants, unfortunately, was not investigated.2.
- Developing the model the parameters of absorption of heavy metals by the biofilter, which were obtained experimentally, were used. Also, in the modeling, an assumption was made that the mass of moss in the biofilter does not change, and also that the moss uniformly absorbs heavy metals. The concentration of heavy metals in the air was calculated according to formula (1), the coefficients for which are taken from the study [21]
- Given the limitations of the proposed methodology, the results for all pollutants are proportional. The problem of biofilters location for all pollutants will have the same solution. The danger of Lead for health and relevance for the Republic of Kazakhstan is confirmed by research [20,25]
- Minor corrections have been made
Reviewer 3 Report
I think the article is interesting but not original as there are many studies that are being carried out looking for ways to purify the air. It correctly presents another approach on how to purify the air. I think your contribution is interesting and can help other scientists working in the same field.
Author Response
The article has been amended in accordance with the provided comments of all reviewers. We hope that our contribution is interesting and can help other scientists working in the field of purifying the air
Reviewer 4 Report
General comments:
The manuscript entitled “Reducing Outdoor Air Pollutants through Moss-Based Bio-Technological Purification Filter in Kazakhstan” is submitted to the journal “Urban Science”. The main contribution of the research is the setup of 227 filter sensors over the city, and demonstrating the total number of filters related to the purification of the urban area. The work can be a good reference for the policymaker. However, some minor corrections need to be made before it can be considered for publication.
Specific comments:
1. Line 318-319: Regarding the reason for setting up the most sensors in Almaty, any literature to support the statement?
2. Line 320-323: Please show the total number of populations of each city for comparison.
3. Line 327-329: The author is suggested to show the formula applied for the estimations of heavy metal concentrations.
4. Line 351-352: Why the temperature regime in vivo and in vitro is different? And why compare the both?
5. Line 357-358: Again, why was the experiment set up at these two different location? How the degree of accumulation of heavy metals can be determined by this method?
6. Line 359-360: Will the temperature changes and humidity influence the efficiency of the heavy metal collection?
7. Line 460-462: The table showed only Pb, how about the rest of the pollutant? In this work, is Pb the most harmful pollutant in the city?
Author Response
Thank you for your valid comments, which significantly improved our research.
- We have added the study of air quality in the cities of the Republic of Kazakhstan [20].
- We have added a column with the number of populations in Table 1
- We have added formula (1) for calculating estimates of heavy metal concentrations.
- Stated the reasons for choosing the in vitro regimen. Comparison of regimes is needed to assess the efficiency of absorption of pollutants by moss under different weather conditions.
5, 6. The conducted experiment showed that Sphagnopsida absorbs heavy metals with the same efficiency both in the open air and in the middle of the biotechnological filter. The stability of the temperature regime does not significantly increase the absorption of heavy metals. The obtained results make it possible to focus on ensuring the survival of moss in a biotechnological filter.
- Given the limitations of the proposed methodology, the results for all pollutants are proportional. The problem of biofilters location for all pollutants will have the same solution. The danger of Lead for health and relevance for the Republic of Kazakhstan is confirmed by research [20,25]
Round 2
Reviewer 1 Report
The new version of the manuscript has been sufficiently improved to warrant publication in Urban Science
Reviewer 2 Report
When the device showed the effectiveness, the blank concentrations of moss without exposure to air pollutants should be required.